# Interfacial Effects and the Nano-Scale Disruption in Adsorbed-Layer of Acrylate Polymer-Tween 80 Fabricated Steroid-Bearing Emulsions: A Rheological Study of Supramolecular Materials

**DOI:** 10.3390/nano11061612

**Published:** 2021-06-19

**Authors:** Nana Adu-Gyamfi, Dipak K. Sarker

**Affiliations:** Interfacial Nanotechnology Group, School of Applied Sciences, The University of Brighton, Lewes Road, Moulsecoomb, Brighton BN2 4GJ, UK; dipak.sarker@rocketmail.com

**Keywords:** acrylate, complexation, coadsorption, viscosity, dispersion

## Abstract

The effect of polymer adsorption on the stability and viable shelf life of 55 μm diameter oil-in-water (O/W) emulsions containing the steroid, betamethasone 21-phosphate was investigated. Two acrylate polymers, Carbopol^®^ 971P and 974P, were added in the role of emulsion stabilizers to a model system, representing a non-ionic low molecular weight surfactant-stabilized emulsion (topically applied medicinal cream). For the purposes of this study the dosage of the viscosifier was maintained below 1% *w/v* and consequently, the consistency of the emulsion was measured in the diluted form. One of the polymers was responsible for elevated degrees of droplet creaming and coalescence and this was closely linked to its surface tension lowering capacity. This lowering was seen at 62 mN/m compared to the routine values at equivalent concentrations of 68 mN/m and 35 mN/m for the betamethasone drug and non-ionic surfactant-Tween 80, respectively. The same polymer also demonstrated a predisposition to form low-micron and greater sized aggregates of nanoparticles that led to extensive flocculation and the formation of a sedimentary precipitate, formed from an amalgam of the components found in the creamed droplet layer.

## 1. Introduction

Industrial, food, cosmetic and pharmaceutical emulsions usually consist of mixtures of an aqueous phase with various oils and/or waxes, with droplets stabilized by a layer of surface-active materials. Oil-in-water (O/W) emulsions consist of oil droplets, which are dispersed (dispersed phase) throughout the aqueous phase. An O/W emulsion can be used as a convenient means of orally or dermatologically administering water-insoluble drugs or therapeutics species. Emulsions are widely used for external medical application and for cosmetic applications. Semisolid emulsions are termed creams and more fluid preparations are either referred to as lotions or if intended to be massaged into the skin as lineaments. Oil-in-water emulsions are used for the topical application of therapeutic species, mainly for a local effect. Two-phase systems, such as O/W emulsions, with size ranges engineered for the route of delivery [1,2,3], are used more generally for the delivery of lipophilic or hydrophobic (high log *p*) drugs. Given the interfacial structuring that takes place under these conditions the hydrophobic components, such as drugs are found to accumulate at the O/W interface as studied by electron paramagnetic resonance (EPR) spectroscopy, which probes the unpaired electrons in radicals and complexes [2]. Most common emulsions (coarse dispersions) have a particle size at the upper end of the colloidal size range that is with a diameter in the 1–10 μm range [4]. Although particle diameters as small as 0.1 μm (100 nm) and as large as 100 μm are not uncommon in some preparations [3,5]. This particle size and the specific coating of the droplets have a significant bearing on the longer-term stability of numerous commercial medical and non-medical products. Medical products based on nanoparticle and micrometer-scale oil droplets include therapeutic microemulsions (oil-carrying micelles), topical emulsions or creams for inflammatory skin disorders and viral infections [6,7,8,9]. Topical microemulsions, often referred to as swollen micelles [1], are used for the treatment of yeast infections incorporating both Tween 20 and clove oil (eugenol alcohol) formulations to give oil-based nanoparticles with a hydrodynamic diameter (diameter) of 34–50 nm [6]. However, on a larger scale the surface structuring of hydrophobic compounds (e.g., drugs, higher alcohols, flavors, perfumes, pesticides, etc.) and liquid crystalline phases formed at the O/W interface were found to give the emulsion a higher degree of relative stability [7]. In addition, a mixed interfacial composition of low molecular weight emulsifiers, block-copolymer surfactants and hydrophobic compounds aids the formation of small and mechanically robust oil droplets. The structure of emulsions versus the concentration and surface loading of the emulsifier Tween 80 was also found to dictate droplet size profiles [8,9]. In addition, the highly used targeted approach to drug delivery, state-of-the-art and specific physico-chemical characteristics of nanocapsules and microcarriers based on interfacially-deposited layers of polymer for a range of polymers, such as acrylates, poloxamers and polylactic acid and for the encapsulation of therapeutic moieties including eugenol, indomethacin, dexamethasone, oestradiol and diclofenac are masterfully described in work by Fessi et al. [10] and Mora-Huertas et al. [11].

The term stability has distinct meaning in the context of coarse dispersions and refers to dispersion homogeneity. In many practical situations it is taken to mean that there are no visual and immediately recognizable signs of phase separation (splitting; syneresis) over a period of time. A suitable stability is where the dispersed globules retain their initial character and remain uniformly distributed throughout the continuous phase. This term is also used in another context to mean that the particles have no tendency to aggregate, so if particles in the dispersed form show a tendency to sediment or cause splitting and phase separation leading to ‘creaming’ over a period of storage, that dispersion would be termed unstable. A stable dispersion of a drug is therefore, one in which particles resist flocculation or aggregation and exhibit a longer shelf life and this will in turn depend upon the balance of the repulsive and attractive forces (as described by Derjaguin-Landau-Verwey-Overbeek (DLVO) colloid and particle stability theory) that exist between particles and the steric separation of the deformable fluid droplets due to absorbed surface-active agent (surfactant). The shelf life of a product is defined as the time the product will remain in satisfactory form (maintaining at least 90% of its active nominal drug content) when stored under the normal range of directed storage conditions [1,3,4]. The aim of most formulated product manufacturers (lubricants, foods, medicines, etc.) is to maintain the integrity of physical presentation and thus drug or active chemical agent efficacy by sustaining the shelf life of the product.

An emulsion is a coarse metastable form of colloidal dispersion even where emulsifiers and stabilizers are used. For any emulsified system both the active ingredient (active) and excipients must be compatible. The physical stability of an emulsion is determined by well-elucidated phenomena such as coalescence (droplet fusion), creaming (flotation of oil droplets to form a concentrated oil-rich supernatant layer), Oswald ripening and flocculation [4,5]. Creaming or widespread droplet buoyancy is one of the main factors that affects the appearance of products, since bigger droplets and close proximity can increase the rate and extent of droplet growth via both coalescence and reconstitution (Oswald ripening) at the expense of smaller droplets and thus phase separation in the product. Creaming can be significantly slowed by producing an emulsion of small droplet size, by increasing the viscosity of the continuous phase, by reducing the density difference between the two phases and by control of dispersed phase concentration. The manufacturing process during emulsification, includes surfactants with specific geometries and wetting characteristics and in particular when combined with the imparted high-shear mixing of oil and water phases, under conditions of dispersed phase fluidity, proves to be the biggest determinant of the average size of droplets found in the product. The viscosity of the continuous phase of a product is varied when a product is stored under low temperature (increased) or higher temperatures (decreased). For example, at lower temperatures this also reduces the kinetic energy of dispersed particles, which in turn decreases the rate of Brownian motion of the dispersed oil phase droplets and their internal and external supramolecular ‘cargo’. Creaming can also be slowed significantly if the densities of the two phases (particle and surrounding medium) are more closely matched, possibly due to adsorption of material on the oil droplet surface [12,13], which might raise the unit density over that of pure oils of, 0.80–0.87 g/cm^3^ for mineral oils and 0.91–0.93 g/cm^3^ for many vegetable oils at ambient temperature.

In certain circumstances, the particles in a colloidal dispersion may adhere to one another to form aggregates and supramolecular structures [14,15] that may settle-out under the influence of gravity, as evidenced in previous studies [16,17]. This flocculation process is the aggregation of droplets in loosely-bound clusters within the emulsion, where individual droplets retain their discrete identities [13] but all the conjugated clusters behave as a single unit, which in turn may increase the rate of creaming at room temperature (25 °C). Flocculation occurs because the oil droplets are surrounded by protective sheath of emulsifying agents (surfactant; wetting agent; surface-active material) and droplets interact with one another with the dispersing agent acting as an intermediary. However, coadsorption and mutual adsorption of surface-active solids [16] and particles [18,19,20,21,22] on adjacent droplet surfaces and displacement of the primary surfactant can cause instability through a complex process known as competitive displacement [19,22]. The disruption or re-enforcement of these consistent lateral interactions in the surface-adsorbed layer has been the focus of many mechanistic studies [19,21,22].

Medical grade carbomers (carbomers USP/NF; Carbopols^®^) are synthetic high molecular weight complex cross-linked polymers [23], based on the monomeric unit of acrylic acid (CH_2_=CHCOOH), also commonly known as carbovinyl polymers. The basic unit of the polymer has the overall empirical structure of sugar-(C_3_H_4_O_2_)_n_, with most molecules having a log *p* value of around 0.2–0.3 but that is dependent on the degree of cross-linking and shielding or removal of polar functional groups [23]. The degree of cross-linking is also thought to alter the p*K*_a_ of the polymer, with values ranging from 5.3 to 6.7 for linear and more highly cross-linked polymers, respectively. The extent of cross-linking is varied between different samples and varied between their intended uses as pharmaceutical excipients. Likewise, the viscosity of samples in aquo also varies as a function of molecular cross-link density at room temperature. The spacing between crosslinks for the medical grade polymers, is 237.6 kDa and 104.4 kDa for Carbopol^®^ 971P NF (sparsely cross-linked) and Carbopol^®^ 974P NF, respectively. The two variants, Carbopol^®^ 971P (sparsely cross-linked) and Carbopol^®^ 974P (highly cross-linked) have molecular weights of 1.25 × 10^6^ Da and 3 × 10^6^ Da, respectively [23] and dynamic viscosities at 0.5% *w/v* concentration in water of 4000–11,000 mPas and 29,400–39,400 mPas, respectively. The same authors describe the structure of 971P and 974P molecules as ‘loose fishnet’ (closed but more ionizable groups) and ‘fuzzball’ (open but less ionizable groups) geometric structures, respectively. The carboxylate content and allyl saccharide (sugar) content of 971 and 974 carbopols is approximately 56–68 and 0.75–2%, respectively.

Carbopols are mainly used in liquid and semisolid pharmaceutical formulations, as bioadhesives [24], controlled release reagents, suspending or viscosity-increasing agents [23], as such they are used in medical creams, gels and ointments for ophthalmic, mucoadhesive vaginal or rectal applications [24] and more general topical applications and additionally, in buccal (as mucoadhesive tablets) and oral solid dosage forms (tablet coatings). Given their widespread and universal use, including their gelation and water-binding capability, they are of real interest for the specific role they play within complex mixtures and emulsion formulations. A dilute solution of carbomer used in solution form is very weakly acidic (pH 4–6) but is usually formulated in a highly buffered environment. Our work evaluates the role commercial texturizing agents can have on the shelf life or longer-term stability and suitability of model liquid paraffin-based low phase volume emulsions. Other commonly used emulsifying polymers such as Pluronics^®^ [17] and low molecular weight simple emulsifiers, such as Spans^®^ and Tweens^®^ [21,22] stabilize emulsions against coalescence by diffusing rapidly from bulk-to-interface and within the plane of the interfacial adsorbed layer forming a “protective skin”, thus providing an effective steric barrier to droplet fusion and thereby act by retarding droplet coalescence [12]. As a result, they also have added-value in consumed products since they are non-toxic and, therefore, used ubiquitously.

The steroid (cortico-steroid) class of molecules includes well-known anti-inflammatory therapeutics, such as hydrocortisone, prednisolone, betamethasone [25] and dexamethasone in addition to sex hormones, such as progesterone and testosterone. All are based on a cholestane tetracyclic triterpene structure with generic skeleton structure of 27 carbon atoms with obvious chemical similarity to their biological precursor, cholesterol. Betamethasone can be used conveniently in molecular studies as an analogue to represent a class of biomolecule, drug and planar organic molecule because of its cholestane molecular skeleton and World Health Organisation (WHO) status as an essential medicine. Betamethasone (C_22_H_29_FO_5_; MW 392; log *p* 1.9) is often formulated in emulsion form as a steroidal cream (0.1% *w*/*v*) in many commercial preparations used to treat psoriasis or dermatitis by reducing inflammation of the skin. The particle size of individual droplets and consistency of the formulation influences ‘spreadability’ and more significantly, penetration of the drug into the subcutaneous tissue. In standard generic formulations of this type, ingredients include the phosphate, dipropionic acid or valeric acid (CH_3_(CH_2_)_3_COOH) derivatized cortico-steroid, betamethasone BP 0.1% *w/v* (active or drug), preservative (chlorocresol, 0.1% *w*/*v*), dispersed phase (liquid paraffin at 30% *w*/*v*), emulsifier blend (cetyl alcohol 8.5% *w*/*v*, emulsifying wax (a mix of polymers, such as cetomacrogol 1000, at 1.5% *w*/*v*) and propylene glycol at 1% *w*/*v*) and buffer salt (phosphates and citrate at 0.1% *w*/*v*) solution. 

For the purposes of comparison, the effect of polymers and primary emulsifiers, a simple model template emulsion with a target droplet diameter of approximately 50–60 μm with simple emulsifier, Tween 80 and preservative sodium azide was selected as the assessment platform. This emulsion format was selected as a standard coarse-structured emulsion because of the ease of assessment of the influence of additional additives and assessment of the formation of molecular nanoarchitecture and supramolecular structure. The aim of our work is therefore, to assess the suitability of mixtures of surface constrained and bulk concentration of ‘active’ excipients and to investigate the stability factors and extent of influence on the composition of non-sterile oil droplet-based nanostructured O/W dispersions [26,27]. Accelerated stability studies were adopted (including centrifugation and temperature modification), which were designed to increase the rate of physical change in the emulsion system, using exaggerated storage conditions and modification of molecular interaction as part of the formal evaluation. These environmental changes were envisaged to aid the elucidation of droplet surface characteristics. Data from these studies allow long-term chemical effects under non-accelerated conditions to be assessed and also provide an idea of the effect of short-term excursions outside the label (optimal) stipulations that might occur during transportation of the class of products concerned. A series of planned experimental regimes using an array of analytical methodologies were used to assess the mechanisms that influence product structure and persistence. 

## 2. Experimental

### 2.1. Materials

All fine chemicals had a purity of greater than 98%, except where identified. All solutions and suspension fabricated used surface chemically pure filtered and deionized water with a resistance of >18 MΩ/cm (surface tension (γ) 72.8 mN/m at 20 °C) and routinely used as a solvent for 50 mM sodium phosphate buffered at pH 7.0 (sodium phosphate buffer salts were obtained from BDH, Poole, UK). Buffered solutions were routinely found to have a surface tension of at least 72 mN/m at 20 °C. The temperature where standard ambient conditions are reported was 25 °C unless stated otherwise. The drug sample (hydrophobe; log *p* 1.8) incorporated into all emulsions was betamethasone 21-phosphate, sodium salt 97% (BMP; cortico-steroidal drug; type B7652, lot 055K1190, Sigma, Gillingham, UK; MW 516 Da) for occasional tensiometric comparative purposes betamethasone 17-valerate 99.7% (BMV; Vetranal^®^) was used (cortico-steroidal drug; type 46074, lot 2123X, Riedel-de-Haen, Munich, Germany; MW 477 Da; log *p* 3.7). Commercial-grade thickening homopolymers (Carbomer^®^) that were added to formulations included Carbopol 971P NF, lot CC94AAT045 (homopolymer type A) and Carbopol 974P NF, lot CC91AAB542 (homopolymer type B), Noveon (BP Goodrich), Cleveland, OH, USA. The molecular weights of these polymers [23] were taken to be polydispersed (intermediate viscosity grades at 0.5% *w*/*v*) and thus relatively difficult to quantify exactly. Concentrations were thus estimated on a weight per volume basis. The carbopols are polymers of acrylic acid, cross-linked with allyl ethers of pentaerythritol and approved for pharmaceutical use by the US Food and Drug Administration (FDA; US Pharmacopoeial Standard USP 29-NF 24). A concentration of 1 mM of Tween 80 (P1754, MW 1.3 kDa, Atlas Chemicals, Widnes, UK) was used as the primary emulsifier in the manufacture of all emulsions. All oil-in-water emulsions were also prepared at a phase volume (φ) of 0.4, using chemically stripped fresh 0.85 g/cm^3^ density light liquid paraffin (Fisher Scientific, Loughborough, UK; batch 0422358) as the dispersed phase. The density of water and buffer used in experiments is approximately 1.00 g/cm^3^ at 25 °C and 0.99 g/cm^3^ at 40 °C. The potent, and simple antimicrobial preservative sodium azide (S8032, lot 27H2606, Sigma-Aldrich, Gillingham, UK) was included in emulsions stored over several days at a concentration of 10 μM. At this concentration of preservative there is no significant influence on the surface composition of oil particles or any indication of significant presence at the air–water (oil–water) interface.

### 2.2. Methodologies

#### 2.2.1. Drug and Drug-Bearing Emulsion

To make the emulsions, aqueous stock solutions of water-dispersible emulsifier (surfactant or polymer) and components of interest were formulated with care avoiding contamination. All solutions were prepared in 50 mM sodium phosphate buffer, pH 7.0 and at 25 °C. Batches of emulsions were formed from solutions and paraffin oil (ratio 60:40) sheared at significant and reproducible speed with a laboratory emulsifier-mixer (Silverson Machines Ltd., Chesham, UK, serial No. 17761) for 5 min on the highest setting (equivalent to 20,000 rpm). Betamethasone phosphate was loaded into emulsions at a concentration of 0.65 mM (0.0335% *w*/*v*) and the Tween 80 concentration was 1 mM. The phosphate derivative of the drug was used primarily because of its more favorable water solubility at ambient temperature and lower tensiometric response.

#### 2.2.2. Emulsion Texture and Stability

The samples were exposed to temperatures of 4 °C, 25 °C and 35 °C to test the thermal stability of the coarse dispersion exposed to a range of realistic storage environments. Stability analogous to a shelf life evaluation was assessed by both rheology and particle-sizing in addition to visual assessment. The stability of emulsions was, therefore, evaluated by measuring the ratio of visible creamed layer height from 300 cm^3^ of emulsion to that of the total height of the sample, for each condition and time period examined in identical storage vessels. Almost immediately after mixing, given the method of preparation, two layers formed, although most of the mixture remained emulsified, with an upper portion of coarse droplets ranging from 3 to 100 μm (or larger) in diameter but with an average of 50–60 μm, whilst there is a nano-sized and low micron-sized droplet phase with particles of 100 nm to between 3 and 1.5 μm in the lower portion. To evaluate the extent of this separation over time the percentage of creaming was measured by taking the ratio of pure emulsion phase height (ignoring the creamed layer) to the total initial sample height [27]. 

The dynamic viscosity (η) and viscoelastic characteristics (complex viscosity, η*; complex modulus, *G**) of thoroughly homogeneous 1 cm^3^ samples was measured using a Rheostress 1 Rheometer (Thermo-Haake, Karlsruhe, Germany) and Rheo-Win software at 25 °C [17]. Changes to the product texture can be identified by both viscosity and viscoelastic rheological measurements on semisolid samples. The acquisition procedures of both types of rheological data were identical, with only the instrument parameters being subject to alteration. The instrument was set-up for use with a 3.5 cm diameter cone with a 1° angle (C35/1 bob), relevant stationary base plate (MPC35) and 0.054 mm gap between the two parts, with the sample placed in the centre of the base plate [17]. The angular frequency range for creep measurements was consistently set to 0.1–1.0 rad/s. The apparent viscosity (η) was estimated as an average of 3–4 values between shear rates of 30/s and 90/s (with median or midpoint value typically at a shear rate of 60/s) over a shear rate scan from 0/s to 300/s. The viscosity values around the median value were universally found to lie within ±5–10% of the median value. For viscoelasticity measurements (complex modulus and complex viscosity) a constant frequency of 1 Hz and imposed strain of 0.05–20 Pa, where used to follow the viscoelastic properties of the sample. The respective modular and viscosity values were all recorded at an imposed stress of 440–450 mPa. Each test was repeated in triplicate to verify the consistency of results. The magnitude of the response can be compared with pure water, which has a dynamic viscosity of 1 mPas and a complex modulus of less than 1 mPa. A viscosity of less than 20 mPas for a sample suggests it has characteristics of a ‘water-like’ solution. Visible changes in the consistency of solutions are detectable above complex viscosity values of 300 mPas.

#### 2.2.3. Particle Size, Charge and Stability Measurements

The size of droplets and a relative change in droplet size can be used to chart coalescence and flocculation processes within the population of dispersed emulsion droplets. The sizing of micron-sized droplets was undertaken on a Mastersizer MS20 (Malvern Instruments, Malvern, UK) [12]. The optical presentation was set to the appropriate position for oil-in-water emulsions. Measurement was initiated with a purging step where 80 cm^3^ of pure filtered water was inserted into the circulator with stirring at 75% of the maximum speed. The probing laser beam was then aligned to pass through the sample in preparation for measurement. The sample container was inverted a couple of times to redisperse the emulsion and then 1.0 cm^3^ of the emulsion was taken from the centre of the container and added to the circulator running at high speed. Small drops were added until the obscuration of the light beam was in the ideal range. Having accumulated duplicate data, the software then calculated the volume-based droplet diameter (*D*_4,3_). This process was repeated in triplicate for all samples, with ten readings being averaged for each independently fabricated formulation. Averages and therefore the error bars representing the standard deviation were therefore based on three independent samples. Particle sizing was undertaken at various time periods after emulsification. 

For occasional submicron range particle and aggregate sizing, a Zetasizer^®^ Nano ZS90 (model ZEN3690, serial No. MAL 1057082), Malvern Instruments, Malvern, UK was used for Sauter average hydrodynamic diameter (*D*_3,2_) size measurements in the range 0.3–3000 nm. As part of the dynamic light scattering technique, the Zetasizer provided the *z*-average value of the particle sizes of the particulates within the sample [17,27]. Set to solvent refractive index 1.33, solvent viscosity 1.0 mPas, dispersant dielectric constant of 78.5, measurement temperature 25 °C, polydispersity was customarily in the range 0.5–1.0 and the count rate per second was typically 30–40 kcps. Estimates of size were based on an average of three independent sample measurement scans (*n* = 3) with an average based on 10 individual counts. The scattering angle was set to 90° from a laser tracking system of 0.1–1 mW power and wavelength of 830 nm. Particle hydrodynamic diameter was measured using an approximation of Mie theory and the Stokes–Einstein equations written into the software. Particle surface charge (ζ-potential) was measured via the application of a variable potential (±5 volts; ±5 mA current) and Smoluchowski approximation across three scans for each of the three independent samples (*n* = 3), which was used for provision of the ζ-potential using the in-built anemometer [17]. All samples were ultra-sonicated for 3 min and filtered through a 0.45 μm PTFE porous filter units prior to measurement. Sizing used a 1 cm path length four clear-windowed plastic cuvette (CXA-110-005J UV grade cuvette, lot: 11388773), Fisher Scientific, Loughborough, UK [13]. The ζ-potential measurements required use of a polycarbonate folded capillary cell (product DTS10T0) from Malvern Instruments, Malvern, UK [14].

#### 2.2.4. Models of the Interface Using Tensiometry

The surface tension was measured using the Wilhelmy plate method, which involved measuring the weight of an immersed glass plate (2 cm × 2 cm × 0.01 cm) in a thermally equilibrated solution with a transducer. The surface tension of the air–water interface was calculated by using an MS Excel spreadsheet written to measure dynamic surface tension based on appropriate calibration. Whenever possible, the measurement was performed in triplicate on independent samples (*n* = 3) and the average value was used. The custom-built apparatus was constructed from a combination of modular devices and software. These include a PowerLab ML826 analog-digital 2/26 converter (serial No. 226-0075), MacLab bridge amp (No. 2623) and PanLab (TR1202PAD) isomeric transducer (No. 727509) and Chart 5 (2006) software, AD Instruments, Australia. Measurements of equilibrium surface tension (γ_eq_) were made at 25 °C following a 15 min period of thermal equilibration [13,28,29]. The air–water (A/W) surface tension was used as an indicator and analogy of the likely oil–water (O/W) interfacial tension. Although not directly comparable there is considerable and predictable similarity between the two measurements undertaken under standard conditions. For reference purposes a potent surface-active agent sample, such as sodium dodecyl sulfate (SDS) at 1 mM concentration has an equilibrium air–water surface tension (γ_eq_) of 25 mN/m and the equivalent value for pure water is >72 mN/m at 25 °C [20].

#### 2.2.5. Microscopic Imaging of Oil Droplets and Nanoparticles

Oil droplet flocculation images undertaken in triplicate (*n* = 3) were captured with ×1 or ×2 zoom with a 3 mm aperture camera (picture quality 12 megapixels/field of view) linked to a Nikon Eclipse E200 MV RS (serial number: 105954) binocular light microscope (Japan) with a magnification of 40 times normal in a similar manner to the method of Hiranphinyophat et al. [27]. Sizes were estimated by comparison (*n* = 3) with a calibrated 25 μm gold-plated tungsten wire (Luma Metall, Kalmar, Sweden) at 25 °C. The magnification of sample for measurement was provided by a ×40/NA0.65 Plan (∞/0.17 WD 0.65) detachable lens also from Nikon. The morphology and sizing of surfactant and polymer nanoparticles was studied using a high-resolution scanning electron microscope (SEM). Aqueous samples were diluted 10-fold with distilled water and of 100 μL having been 0.22 μm filtered were deposited on top of smoothed double-sided carbon tape (Agar Scientific, Ltd., Stansted, UK) mounted onto a 1.2 cm aluminum stud, and air-dried in a desiccation chamber for 48 h. The dried samples were treated with a 2 nm surface coating of palladium to aid imaging. Equilibrated samples were then loaded under high vacuum into the SEM holder ready for measurement [15,17]. SEM images were recorded on a Jeol, JSM 5310, (Tokyo, Japan) scanning electron microscope, with an operating acceleration voltage of up to 25 kV [29]. 

## 3. Results and Discussion

### 3.1. The Model System and Control

A typical surface tension plot for Tween 80 surfactant is shown in Figure 1 and indicates a notable decrease in surface tension on the incremental increase in concentration, followed by a lower plateau established at 35 mN/m. The intersection between the slope and plateau corresponds to a critical concentration point at which micelles start to form (critical micelle concentration; CMC). This CMC was determined to occur at a concentration of 1 × 10^−4^ M (Figure 1), consequently all solutions used to fabricate dispersion in further experimentation contained at least 10^−3^ M Tween 80 (arrowed in the figure) in formulations. In supplementary sizing measurements of Tween 80 micelles the diameter of a monodispersed nanoparticles at 6 nm was unaltered (not shown), within the precision of the technique at 1 nm, by the inclusion of betamethasone drug (BMV or BMP) and methodology validated by literature values [13,14]. The critical micelle concentration for Tween 80 is low when considered against the vast surface area of 0.14 m^2^/g for a 1% *w/v* oil droplet dispersion with an average droplet size of 55 ± 30 μm and the coverage needed. The technique can be used more successfully to confirm and chart competitive interfacial processes that change with type and concentration of primary surfactant used. However, it is worth noting that charged surfactants such as sodium dodecyl sulfate (SDS) with a short C_12_ aliphatic tail and compact anionic head group are dissimilar to Tween 80, which has a longer C_18_ alkyl tail as part of its structure but with a more ‘bulky’ and voluminous polar head group [19,20] and that has a lower equilibrium concentration based critical micelle concentration largely on account of the larger apolar surfactant tail (SDS CMC 1 mM). Equilibrium surface tension values that might be seen with compact molecules such as C_12_ SDS (γ_eq_ = 25 mN/m) are much lower than Tween’s due to the efficiency of interfacial loading [19]. This is a function of the capacity of ‘short and streamline’ molecules to form compact structures and lateral firm lateral packing or associations at the interface [26,27]. This is important when considering the chemical nature of adsorbed polymers such as carbopols [23,30,31,32,33,34]. 

The functionality of carbopols when used for gels as suspending agents [23], in interacting with fragrance or hydrophobe molecules [26], non-ionic emulsifiers [31], anionic surfactants [34], drugs and sterane sex hormones such as estrogen [32] and through molecular interactions or micelle forming polyelectrolyte or hydrophilic polymers used for drug delivery and gene therapy of the polyoxazoline class through hydrogen bonding has been investigated [33] at some length. Given the nature of the carbopol molecular skeleton and voluminous polar Tween 80 headgroup, electrostatic or hydrogen-bond like interaction based on the development of partial charges between polymer and surfactant is highly likely [5,13,19,22]. Similarly, interaction of the polar regions of a drug such as ‘betamethasone class’ and hydrophilic polymer or Tween 80 surfactant either in the monomeric form or encapsulated by virtue of its apolarity, within micellar forms, is also highly likely [13,14] and this is yet further reinforced, as was previously reported for entrainment of hydrophobic entities and molecules (log *p* > 1.5) in bilayered vesicular carriers [15,25], as might be expected for BMP with a log *p* value of 1.8.

The effect of droplet density and phase volume on emulsion viscosity shows a consistency of bulk viscosity between phase volume values of 0.3 and 0.45 (Figure 2). Given this plateau range of values, preceding a steady increase, a phase volume value of ϕ = 0.4 was used as the basis for further test sample studies (arrowed in the figure). Using a dispersion with a large number of droplets is more revealing in terms of screening for surface effects, droplet creaming, flocculation and coalescence because effects are somewhat magnified. Numerous historical and recent studies have demonstrated the pivotal role of interfacial rheology, described in its viscosity or elasticity and lack of interfacial mobility in droplet stabilization plays in securing dispersion consistency [2,12,16,19,21,28]. Calibration of the effect of inclusion of betamethasone-21-phosphate (BMP) in emulsions fabricated using 1 mM Tween 80 are shown in Table 1. The data suggests there was no significant or large-scale change on emulsion size and size distribution on inclusion of BMP up to concentrations of 0.65 mM values, thereafter increasing marginally. Consequently, a concentration of 0.65 mM was used in subsequent test systems, as this represented the highest inclusions without apparent effect on droplet size. 

Control emulsions were based on Tween 80, as the primary emulsifier with the inclusion of the steroidal drug, betamethasone-21-phosphate (BMP) at a concentration of 0.65 mM, as an interfacial intercalation species, thought to be able to interdigitate with surfactant alkyl chains and polar head groups [25,26,27]. Intramolecular interaction between the primary adsorbed layer materials and sublayer materials was indicated previously by both neutron reflectometry [34] and EPR spectroscopy [2], used to further investigate the role of carbopols in surface adsorption behavior. The control test samples (*n* = 3) produced emulsions with a minimally changing average droplet diameter of 55–65 μm, from the initial preparation time (time, *t* = 0 min) to 100 min, respectively (Figure 3) under large applied centrifugal forces (10,000× *g*). This type of centrifugal study allows the perikinetic or accelerated ‘shelf life’ assessment of droplet interfacial robustness by forcing droplets together within any ‘creamed’ layer in large volumes. Enhanced initial fusion of droplets was revealed by coalescence shown in perikinetic trials involving the incorporation of 1.5% *w/v* carbopols in the control sample (Figure 3). Emulsions bearing carbopol polymer showed a degree of further resistance to emulsion splitting through widespread coalescence with approximately the same value after 60–100 min of centrifugation with sizes of 100–105 μm and 110–112 μm for carbopol P971 and P974, respectively. Notably these values are roughly 1.6–1.7 times the droplet diameter seen at 100 min of centrifugation for the control emulsion. There is a hint in the figure that P971 may provide a better form of emulsion surface-structuring, since it has the lower droplet size change with time. Here, prolonged applied pressure (under buoyancy influences) through direct contact in the creamed layer across both carbopol systems enhanced coalescence and led to a bigger average droplet size. This suggests that the surface structure where carbopols are involved is either weakened in some manner or depletes the surface of surface-active material via polymer association and aggregation with Tween 80. This and other depletion phenomena have been reported previously in O/W emulsions [12] and aqueous foams [19]. There is of course a possibility for a source of error in the characterization of droplet size due to oil droplet coalescence as reported by Caserta et al. for polymer blends under shear flow [35]. Nevertheless, the size of droplets, lower ambient temperature, relatively short time post manufacture of size measurement and interfacial reinforcement to provide a structurized less fluid adsorbed layer composition tend to support and authenticate the values determined in droplet and aggregate sizing measurements.

### 3.2. Exposure to Harsh Environments and Interfacial Suitability

The base or test emulsion containing Tween 80 and BMP was exposed to temperature extremes to assess the effect of temperature on droplet fluidity, interfacial rigidity and the rate of droplet collisions, all of which can be expected to impact on the extent of droplet coalescence (Figure 4). The data clearly slows little or no real influence at refrigerator temperatures of 4 °C over a period of 10 days on droplet size. However, over the same time period at ambient temperature (25 °C) and elevated temperatures of 35 °C the diameter of droplets increased steadily in a linear fashion as a function of storage time; with the latter, somewhat predictably, changing the most. Emulsions are undoubtedly influenced by temperature in terms of increased droplet average kinetic energy and mobility and an associated decrease in oil phase bulk viscosity and therefore degree of coalesce resistance (Figure 4). The increased extent of coalescence indicated by an average droplet diameter change from the parent control sample at the start of the trial, also points at the nature of the droplet adsorbed layer. Additionally, the emulsifier layer appears to be less mechanically robust on an increase rate and speed of collisions (driven by the temperature augmentation). 

Increasing temperature increases coalescence as revealed by many studies including that of Kerstens et al. [16] by increasing both the movement and the fluidity of droplets leading to fusions that are manifested in larger droplets (Figure 4). Increased emulsifier solubility as the temperature is raised from 25 to 35 °C may also produce a lower interfacial viscosity (and rigidity) as a function of poorer and less dense surface coverage. Stability is also related to bulk effects since increasing the temperature increases the rate of collision via increased translational particle diffusion (*D*) [2]. This is described by a simplified form of the Stokes–Einstein equation [1], with *D* = *kT*/η, where *k* is a constant that relates to the kinetic energy, and also where the temperature (*T*) and the viscosity (η) of the medium (also influenced by temperature) dominate. The factors of particle size, particle number and a geometric factor (which are the same at the start of the storage time) can be removed as relative constants in the relationship. At higher temperatures not only do droplets collide more frequently but also the density difference between dispersed oil droplets and continuous aqueous solution is increased and this may influence the creaming rate according to Stokes’ Law. The density difference between water and oil was 0.097 g/cm^3^ at 25 °C and 0.102 g/cm^3^ at 35 °C with the density of water being around 0.999 g/cm^3^ over this range [1,12]. This effect is analogous to the increased coalescence (and thus droplet size) seen in Figure 3 after 50 min centrifugation when compared to shorter centrifugation times. 

### 3.3. Surface Tension and Interfacial Structure Limitations of Drug and Stabilizer-Emulsifier

The drug used to make the emulsions, BMP, was less surface active than BMV (Figure 5) and all were considerably (60–35%, respectively) less than the 1 mM concentrations of Tween 80 (tension 43 mN/m). An extrapolation for BMP on the plot shows an equilibrium surface tension value of projected to 1 mM of 64 mN/m. It is therefore highly likely Tween 80 dominates the interface based on its ability to lower surface tension (Figure 1) to equilibrium surface tension (γ_eq_) values of 35 mN/m. The surface tension values of carbopol solutions (suspensions) at 20 °C is presented in Figure 6A. Given the same weighed-in concentrations (0.5 g/100 cm^3^) and evident from hand-shaking tests undertaken in parallel, carbopol 971P was able to maintain froth bubbles for longer than 974P. There was some evidence of residue or sediment in the carbopol 974P same at this concentration, which points at polymer coagulation and flocculation. This more persistent formed foam for 971P is possibly due to the presence of more is being present at weighed concentrations of ~0.5–0.6% *w*/*v*, as demonstrated in Figure 6A, with the lower γ_eq_ values of 61 mN/m for 971P, even though the onset of surface tension reduction for 971P as a function of concentration was less pronounced than for 974P. Since a greater quantity of persistent foam was formed with 971 but with 974 polymer-stabilized bubbles being formed but bursting quickly, the likely cause of the rupture was the presence of clearly visible 100 μm “hydrophobic aggregates” (viewed under the light microscope). Aggregates of the 974 polymer in the aqueous phase had a tendency to stick to the glass wall of the sample container, suggesting some degree of apolarity.

Ethylcellulose, a similar homopolymer, was previously reported to form hydrophobic aggregates, similarly the carbomers used may also form such mixed and amorphous micellar aggregates and clusters [36]. Another homopolymer, methylcellulose was previously shown form aggregates and to be highly surface-active and effective as a polymeric emulsifier [12]. Aggregates of carbopol 934 along with the block copolymer, a poloxamer (Pluronic F127) [32], a similar type of acrylate polymer to the two species used in this study and carbopol 971 [33] itself have also been reported in aqueous media systems. Hydrogen-bonding permits the formulation of a complex and aggregates with the unionized carboxylic acid groups generally oriented to the interior and the ionized water miscible groups oriented to the outside of the aggregates [33]. This aggregation in the case of P974 and depletion of active molecule for surface coverage is seen as a plateauing of surface tension values (50–51 mN/m) and diminishing of the related bulk viscosity shown at the higher concentrations of the polymer in Figure 6B. Over concentrations of 0.5–2% *w*/*v* the bulk viscosity of dispersion tested for carbopol 974 was 1–2.5 mPas. This ‘weaker’ texture contrasts with the more homogeneous dispersion of carbopol 971 where a viscosity of 2–17 mPas was observed (Figure 6B). The higher value points toward stronger inter- and intramolecular association with carbopol 971. The difference in viscosity does however, not take account of the residue at the base of the carbopol 974 sample vessel but merely the homogeneous part of the polymer suspension that remained in the dispersed colloidal form.

### 3.4. Bulk Effects Related to Viscosity

The viscosities of the dispersions used to fabricate emulsions at room temperature are presented in Part B of Figure 6. The viscosity of the 971 sample at 2% *w/v* was nearly 10 times that of the 974 even though the 974 had a higher supplier quoted value (4000–1100 mPas and 29,400–39,400 mPas, respectively). This is probably related to the quantity solubilized in the former and the size and extent of cross-linking in the polymer species [23,33]. 971P is a loose “fishnet” format with bands of 237.6 kDa between crosslinks, with an overall molecular mass of 1.25 MDa and this compares to 974P which has the ‘fuzzball’ open structure with 104.4 kDa portions between crosslinks and marginally more than double the overall molecular mass [23]. In any case this represents the bulk conditions of the aqueous phase used to make the emulsions. Mechanical spectra indicate the complex viscosity (η*) of 2% *w/v* solutions (suspensions) to be 18 mPas for carbopol 974 P at 25 °C and a stress of 0.5 Pa, yet the value for 971P under the same conditions gave a value 30 times larger. The gel properties indicate a 2% solution (or suspension) of carbopol were 0.8 Pa for 974 indicating a weak “gel” structure and 3.4 Pa for 971 indicating a gel-like network. The 971 exists as a solution with a particle size of 1.44 μm ± 0.42 and the 974 as a dispersion with fine micrometer particles of 2.43 μm ± 0.94 and visible bigger aggregates estimated to be 1–300 μm. These aggregates may have a significant bearing in the emulsion stability of coarse dispersions [18,26]. This could be in terms of interfacial destabilization as indicated by a bigger average droplet size (Figure 7). In addition, these larger aggregates may also remain excluded from solution and thus the effective solution concentration is lower than that reported. With the concentrations of carbopols (0.5%) incorporated into the emulsions proportionately similar bulk effects may be observed. It is also likely that on the scale of the particles local augmentations in concentration above those seen in the bulk are possible based on the surface tension data, which suggests the polymers have an affinity for the interface (Figure 6A) and form a complex mixture of surfactant and solid particle stabilized Pickering emulsions [18,23,27]. Elsewhere, Pickering emulsions were formed where the interface saw an association between polar nanoparticles, low molecular weight surfactant and polymer to provide enhanced O/W emulsion droplet stability and an association that was pH-responsive, to give a tunable release of associated drug molecules [26].

### 3.5. Coalescence and Interfacial Destabilisation

The effect of the presence of these polymers is shown in (Figure 7A). Here the droplet size is increased as the concentration of carbopol is increased. Two possibilities exist—displacement of the Tween [22], unlikely given the higher surface tensions or domination of the surface by a Tween-polymer complex and aggregate. Bulk viscosities (Figure 7B) are indeed likely to slow-down creaming over the control formed from only Tween 80 with an aqueous phase bulk viscosity of 1 mPas. In line with the aggregate hypothesis, it seems that the polymer aggregates (carbopol 974P) are always present and may increase emulsion droplet size (based on the control). Thus, 974 seems to cause more rapid coalescence and transition from 55 μm to visible oil droplets (0.5 mm) in the emulsion portion and may cause a degree of aggregation of droplets [12,18,37]. This aggregation could be driven by the formation of a Tween 80-carbopol complex and even explain the failure of emulsions over Tween only emulsions as the complex precipitates or is poorly soluble [16,22,38]. This can be evidenced as a precipitated “sludge or residue” that sits at the oil–water (bulk emulsion) interface, which points to the poorer solubility of the 974 polymer. This white layer was always thicker with 974P (2 mm; 4% emulsion volume) than with 971P (1 mm) and was found within 30 min of emulsion preparation. The subnatant in all cases contains smaller emulsion particles and were still evident 5 days after formation of the emulsion, even though showing a gradation of presence from bottom (largely absent) to the top of the subnatant portion witnessed by the 18–46 ± 7 μm diameter of droplets found in the aqueous phase (Figure 8). Charge-screening and its influence on solubility [37] are also likely to lie behind the process of adsorption [38] flocculation or the surfactant and polymer and thereby create a poor surface coverage of oil droplets promoting droplet coalescence [39].

The incorporation of polymers causes less oil to split from the dispersion (Figure 9) with the least emulsion being maintained by the 971P of the two included carbomers it also gave the most subnatant liquid and this is in line with tensiometric data that suggest at the concentrations used (0.5%) that 971 solutions were more surface-active than 974. Carbopol 271P has a molecular weight that is approximately 2.5 times less than that of the 974 [23] and approximately 100 times that of the Tween, this is indicated by the difference in apparent viscosity in Figure 6B and complex module in Figure 7B that should also relate to molecular weight for roughly similar chemical entities to these carbomers. The molecular compatibility of 971 and Tween may have resulted in the greater proportion of split oil and the lower proportion of intact emulsion with carbopol 971P (Figure 9) through less effective droplet coverage. This has significance for formulation of pharmaceuticals and other commercial products, since the polymers are usually added as stabilizers and thickeners but indeed where the primary surfactant/emulsifier concentration is low may indeed act as destabilizers through a number of means, including the nucleation of aqueous film rupture [39]. According to this research it is likely that any supramolecular aggregated microheterogeneous particles formed from polymers with differing surface regions of hydrophilicity and hydrophobicity, their location and thus steric stabilization in the aqueous phase can promote enhanced stability [34] or destabilization [37].

A synergistic association between surfactants, such as SDS and PEGylated polyacrylate polymer was determined by neutron reflectivity measurements but this time at the AW interface [34] and this is the basis of a proposed equivalent interfacial supramolecular structure as also evidenced in the data presented here. Molecular association, manifested as surface elasticity, within and between species constrained at the interface is vital to the resistivity of the droplet surface. In a simple demonstration of the rigidification of an adsorbed layer forged by complex types of molecular interactions, surface elasticity of the O/W interface, as measured by the dilational modulus for simple triblock copolymer (Pluronic F68) was 9–14 mN/m where the equilibrium A/W interfacial tension was 40 mN/m and this indicated a weaker surface structure than found with a strongly interacting surface active protein such as lysozyme, with an O/W dilational modulus of 83–94 mN/m, despite having an A/W interfacial tension value of only 45 mN/m. The possibility for multiple forms of molecular interaction over and beyond hydrogen bonding lies behind the greater adsorbed layer elasticity with the protein as compared to the poloxamer homopolymer [40].

In the case of aqueous interstitial film destabilization [39], it is likely to occur when the particle aggregate has a poor affinity for the aqueous film between two droplets and thus has a high contact angle (and hence poor capillary pressure) of greater than a critical value that for simple models was 70–100°. Here too, a lack of interfacial amphiphilicity is likely to promote coalescence by allowing oil droplets to be linked leading to flocculation and fusion. In a simple measurement undertaken to supplement interfacial tension measurements, with pure droplets of water (0.1 cm^3^) placed on a compacted horizontal mat of carbopol the averaged contact angles (*n* = 3) were found to be 47 ± 2° and 50 ± 3° for 971 and 974 carbopols, respectively. This means that aggregates of polymer are likely to be moderately hydrophobic. Light microscopy images of dispersions that include 50–60 μm emulsions formed from Tween 80, BMP and carbopol 971P, immediately after formation but also the subnatant phase of the emulsion after 10 min standing and a pure sample of carbopol 971P are shown in Figure 10.

The images in Figure 10 strongly support the findings of particle sizing of emulsions (Figure 3) and of carbopols (Figure 7A) along with rheological (Figure 6B and Figure 7B) and foam generation observations, where particles of carbopols were observed. The solid particles observed in light microscopy and measured by light scattering techniques were also clearly observed using the scanning electron microscopy technique. SEM images of emulsifier and polymeric surface-active materials used to fabricate emulsions are shown in Figure 11. Tween 80 particles (micelles) of approximately 5-6 nm and carbopol aggregates of more than 150 nm for 0.1% *w*/*v* 971P samples along with composite larger aggregates of more than 500 nm at higher 2% concentrations of 974P were in support of light microscopy and observational data are also shown clearly by electron microscopy.

Interfacial adsorbed aggregates of 974 by virtue of their location and orientation, could result from a lack of steric screening of hydrophobic regions of the polymer, possibly through complexation [17,38,41]. Essentially, in all ‘surfactant-like’ moieties, to be able to provide necessary interfacial adhesion and persistence the polymer must contain significant stretches and numbers of structural portions of a hydrophobic nature to be surface-active and reside at the O/W interface. However, this very association driven by zonal molecular hydrophobicity could ultimately simultaneously deplete the surface from the more hydrophobic of the two carbopol polymers (Figure 6A). The formation of carbopol-surfactant micelles/aggregates and the ionization of carboxylic acid groups leads to an expansion of polymer chain dimensionality and resultant increase in dispersion viscosity. Intercalation and potential conjoining of the so-called ‘sticky’ zones of polyacrylate by low molecular weight amphiphiles (surfactants) is largely based on the association polar regions of the carbopol polymer [30]. This is thought to be attributable to the direct association of surfactants by a hydrogen-bonding interaction between the individual oxyethylene head groups (of Tween 80 or Pluronic F127) and that of the carboxylic acid groups present in the polyacrylate. A cross-over in the surface tension plots for Tween 80 and Pluronic F127 in the absence and presence of carbopol (934) points to- molecular binding and reinforces the notion of complexation. Through this mechanism unionized carboxylate groups, therefore, permit a greater degree of intermolecular and intramolecular crosslinking through partial charge-induced bonding [31]. According to Barreiro-Igleasias et al., carbopol 934 and pluronic F127 form aggregates and these aggregates were able to bind to the oestradiol drug [32]. Within this solubilization study, the association of surfactant and polymer were thought to involve the multiple mechanisms of hydrogen bonding, apolar association and ionic or electrostatic interaction. Binding of the drug in this case, was found to decrease the release of the drug, although the solubility of the drug was increased with the use of carbopol and surfactant complexation, through an increase in hydrophilicity.

Given the non-ionic nature of the Tween 80 and the ionizable nature of the acrylate polymer with a p*K*_a_ 6.0 ± 0.7, hydrogen bonding and ionization [32,33] between the acrylate polymer and the head group of surface restrained Tween 80 as identified for other systems of polymer and low molecular weight surfactant [13,21,29,34,40] are likely to lie behind any interfacial structuration. A phenomenological cartoon of the O/W interface, multilayer and sublayer structuring in the aqueous phase and supported by other studies [34] effective in the reported work, is presented in Figure 12. A combination of interfacial adsorbed Tween 80 complexed or associated with betamethasone 21-phosphate and simultaneously with carbopol appear to form the basis of stabilized oil-in-water emulsions (Figure 8). The complex network ‘flooded’ with large amounts of ‘cushioning’ associated aqueous solvent appear to be induced by hydrogen bonding, ion-mediated bonding, also possibly using counterions and apolar interaction between the alkyl tails of the primary surfactant. Certainly, the intricacies of the mechanism of involvement are subject to the diversity of potential molecular interactions at the interface. In the scenario presented in Figure 12 a simple homopolymer, with obvious potential of large-scale charge-based or hydrogen-bonding association is seen but this could be exaggerated in the presence of hetero-polymers such as proteins and nucleic acids, often present for example, in drug delivery systems, vaccines and food products.

## 4. Conclusions

Carbopol polymers are simultaneously surface-active yet highly hydrophilic in nature and shown to interfere with the mechanism by which simple surfactants stabilize the interfaces of oil droplets dispersed in water. The increase in the bulk viscosity of the smaller, yet more hydrophilic, open structured and ionizable 971 molecule over the 974 carbomer is also responsible for an increased splitting of the emulsion, although size data indicate a smaller average droplet size for the former at 0.5% *w*/*v* carbopol incorporation. It is likely that both bridging flocculation or depletion flocculation, formation of a tightly-packed creamed layer of particles and oil droplets and interfacial inconsistencies due to malabsorption are responsible for the increased volume of free oil and any emulsion breaking. The nanostructured nature of the components forming the adsorbed layer built on oil droplets with a uniform size distribution can provide the basis for interface studies. The work reveals interesting insights into the role “inert” formulation ingredients can have on the complex pharmaceutical and cosmeceutical products and their shelf stability.

## Figures and Tables

**Figure 1 nanomaterials-11-01612-f001:**
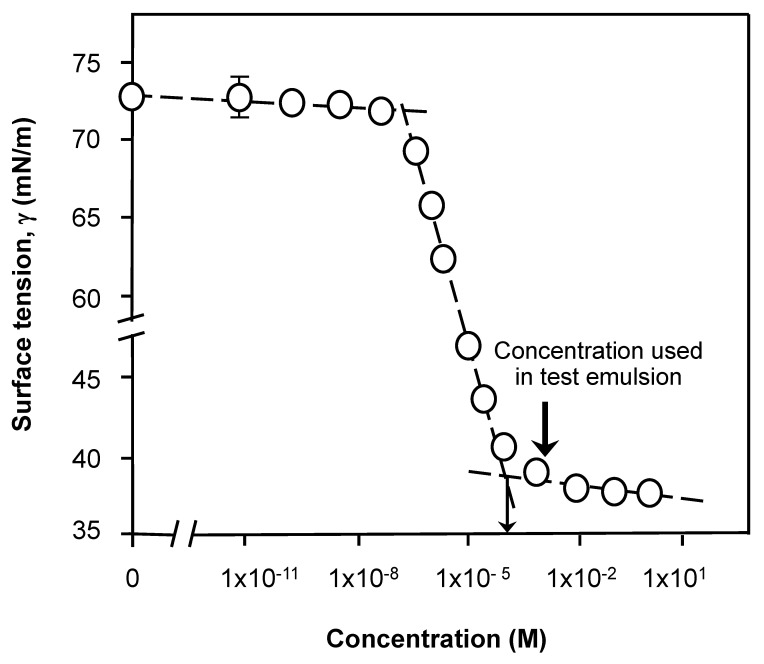
Surface tension for purified Tween 80 solutions at 20 °C as a function of increasing bulk concentration (where error bars are not shown they are of a smaller magnitude than the dimensions of the symbols; lines represent an aid to the eye to show the trend in the data). The arrow in the figure indicates the concentration used in test emulsion. All solutions used for samples were prepared from surface chemically pure 50 mM sodium phosphate buffered water, pH 7.0.

**Figure 2 nanomaterials-11-01612-f002:**
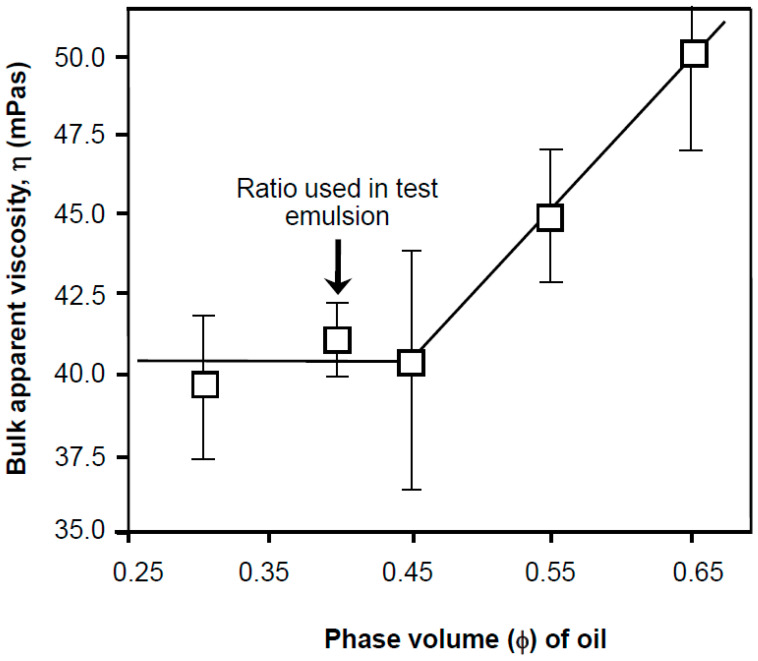
The consistency and texture as viscosity (averaged over a shear rate of 30–90/s) of the base emulsion made from 1 mM Tween 80 and liquid paraffin as the dispersed phase in water in the presence of 1 × 10^−5^ M sodium azide (lines merely represent an aid to the eye to show the trend in the data) as a function of the phase volume of the oil. All solutions used for samples were prepared from surface chemically pure 50 mM sodium phosphate buffered water, pH 7.0.

**Figure 3 nanomaterials-11-01612-f003:**
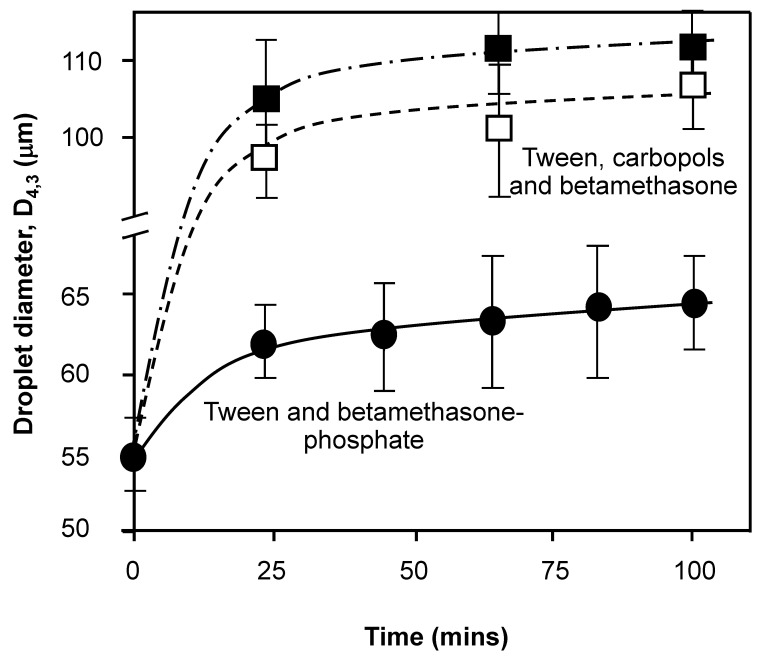
The droplet size of the base emulsion made from 1 mM Tween 80 and liquid paraffin as the dispersed phase in water in the presence of 1 × 10^−5^ M sodium azide (lines represent an aid to the eye to show the trend in the data) as a function of the forced centrifugal coalescence under gravity (10,000× *g*). The samples were prepared from Tween 80 and 0.65 mM betamethasone 21-phosphate (BMP) alone (●) and in the presence of either 1.5% *w/v* carbopol 971P (□) or 974P (■). All solutions used for samples were prepared from surface chemically pure 50 mM sodium phosphate buffered water, pH 7.0.

**Figure 4 nanomaterials-11-01612-f004:**
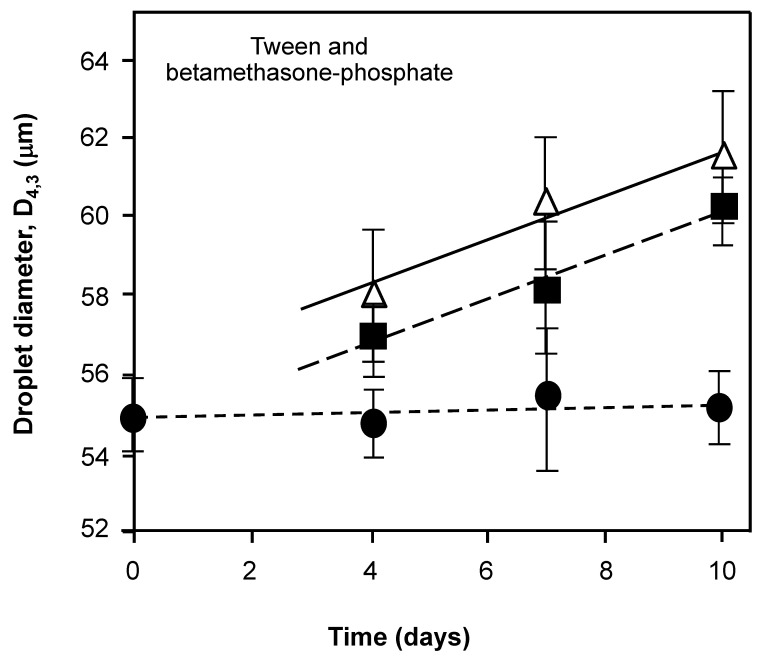
The droplet size of the base emulsion made from 1 mM Tween 80, 0.65 mM betamethasone 21-phosphate (BMP) and liquid paraffin as the dispersed phase in water in the presence of 1 × 10^−5^ M sodium azide (lines represent an aid to the eye to show the trend in the data) as a function of the storage time in days at a range of temperatures. The conditions of storage of the manufactured emulsion were 4 °C (●), 25 °C (■) and 35 °C (△). All solutions used for samples were prepared from surface chemically pure 50 mM sodium phosphate buffered water, pH 7.0.

**Figure 5 nanomaterials-11-01612-f005:**
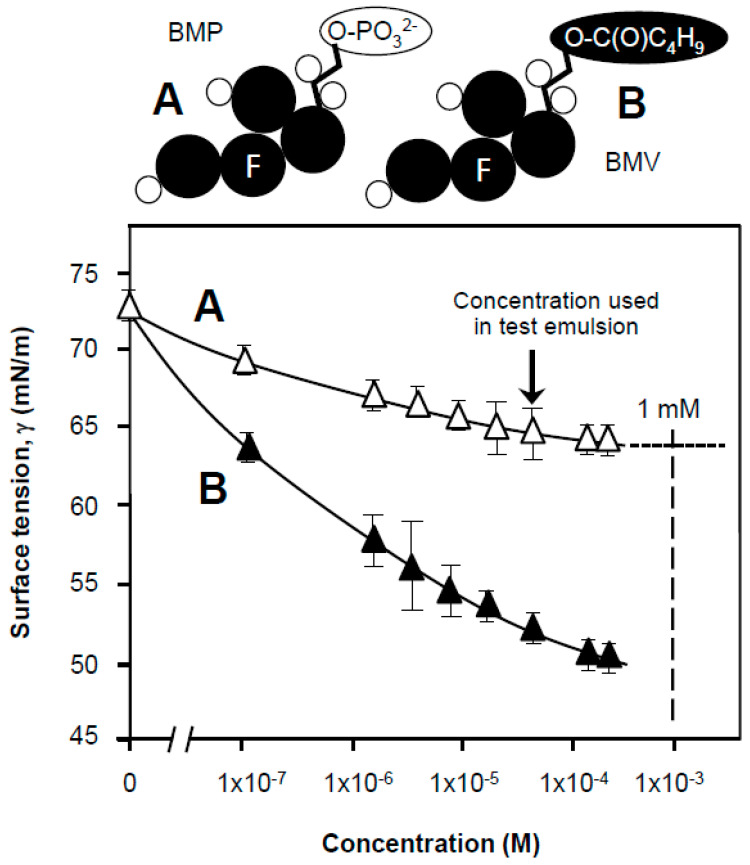
Surface tension for purified drug solutions at 20 °C as a function of increasing bulk concentration (where error bars are not shown they are of a smaller magnitude than the dimensions of the symbols; lines represent an aid to the eye to show the trend in the data). The drug samples were betamethasone 17-valerate (BMV) (▲) and betamethasone 21-phosphate (BMP) as a sodium salt (△). All solutions used for samples were prepared from surface chemically pure 50 mM sodium phosphate buffered water, pH 7.0. In the simplified molecular structures appended to the figure, black coloration is used to indicate hydrophobicity, with “F” representing a fluorinated component.

**Figure 6 nanomaterials-11-01612-f006:**
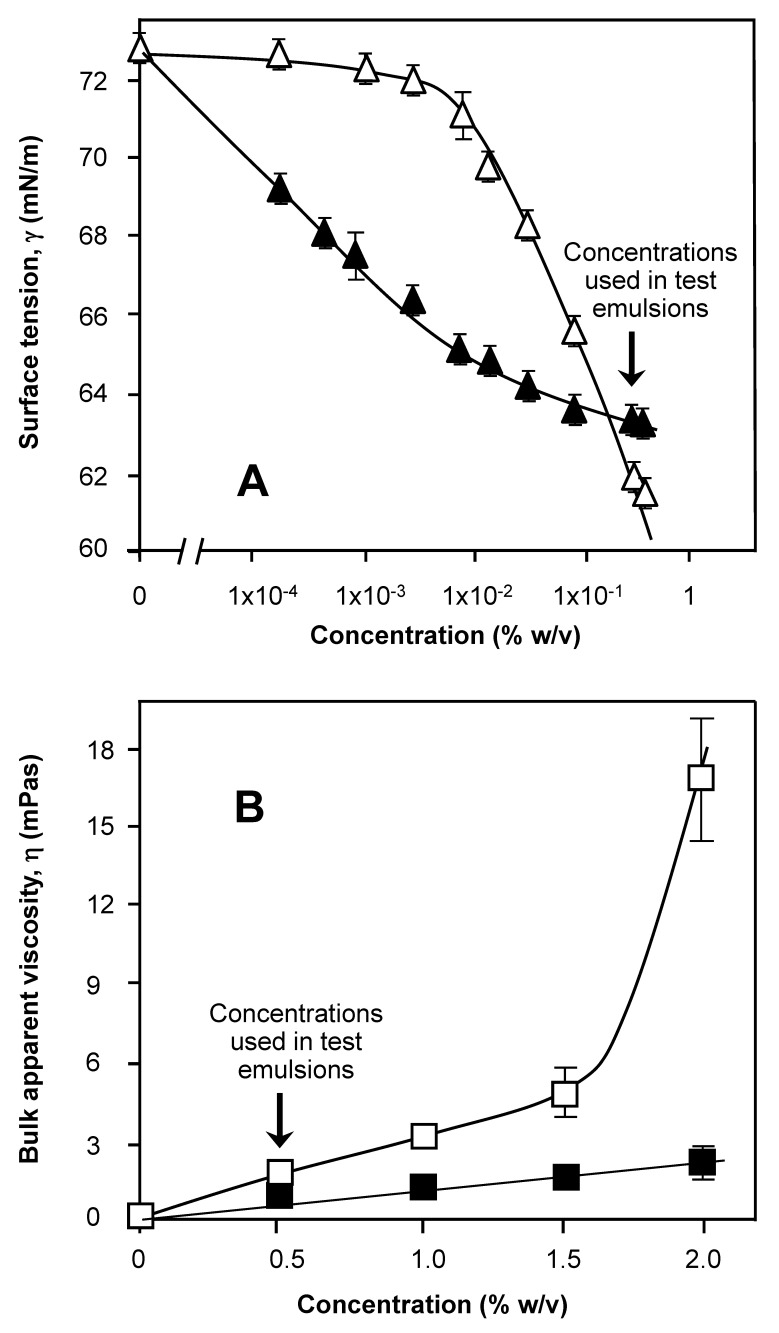
Multiple plot for the test system, indicating surface tension (**A**) and bulk solution viscosity (**B**), estimates for carbopol solutions at 20 °C as a function of increasing bulk concentration, given as a percentage weight-in-volume (% *w*/*v*) in the presence of 1 × 10^−5^ M sodium azide (where error bars are not easily discriminated they are of a smaller magnitude than the dimensions of the symbols; lines represent an aid to the eye to show the trend in the data). The polymer samples were carbopol 971P (△) and carbopol 974P (▲) in tensiometric studies and carbopol 971P (□) and carbopol 974P (■) in rheological studies All solutions used for samples were prepared from surface chemically pure 50 mM sodium phosphate buffered water, pH 7.0.

**Figure 7 nanomaterials-11-01612-f007:**
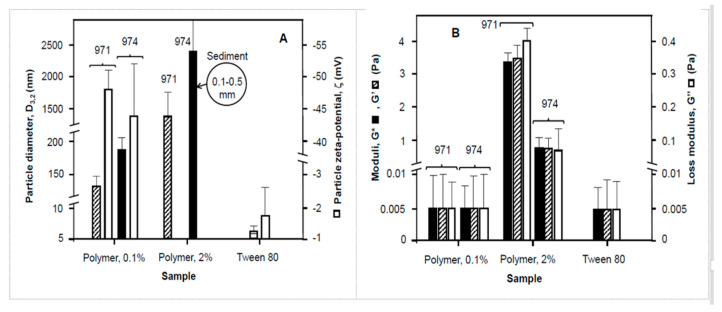
Multiple plot for the carbopol polymer colloidal suspensions, indicating average particle size (solid or hatched bars) and surface charge (unfilled bars) of suspended material (**A**) and also the complex modulus (solid bars), storage modulus (hatched bars), along with the loss modulus (unfilled bars) of suspensions (**B**) at 20 °C as a function of increasing bulk concentration, given as a percentage weight-in-volume (% *w*/*v*). The polymer samples were carbopol 971P and carbopol 974P particle characterization and in rheological studies. Notably the carbopol 974P sample also showed evidence on inspection of limited sedimentation after more than 30 min in the preparation vessel, with particles in the submillimeter size. All solutions used for samples were prepared from surface chemically pure 50 mM sodium phosphate buffered water, pH 7.0.

**Figure 8 nanomaterials-11-01612-f008:**
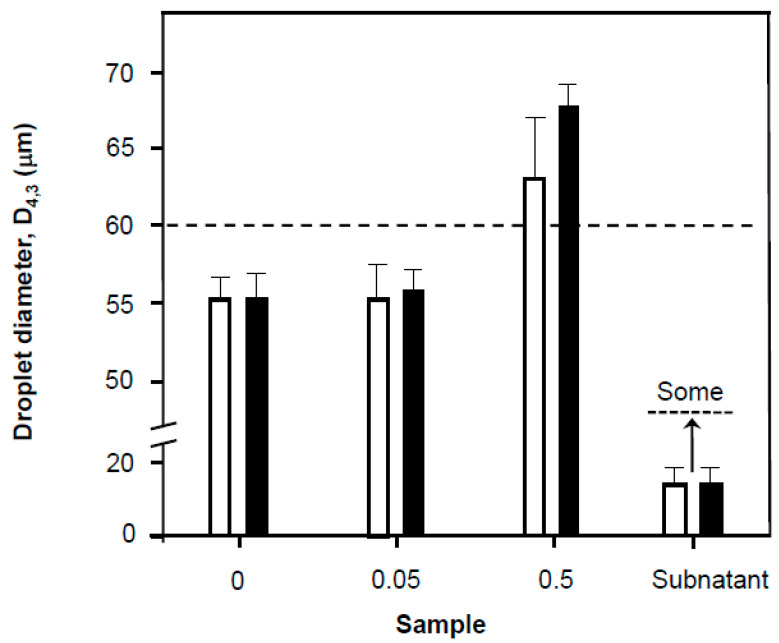
Emulsion droplet size for the test system using 1 mM Tween 80 and 0.65 mM betamethasone 21-phosphate as a function of carbopol inclusion in the aqueous phase at 20 °C also in the presence of 1 × 10^−5^ M sodium azide. The values presented as 0, 0.05 and 0.5 refer to the concentration of included carbopol (% *w*/*v*). The polymer samples were carbopol 971P (unfilled bars) and carbopol 974P (filled bars). The particle size in subnatant aqueous portions of the emulsions formed (control-Tween only and in the presence of carbopol polymers were numerically indistinguishable) are also presented. The broken horizontal line is included to emphasize the difference in droplet diameter at dilute and concentrated carbopol inclusion. All solutions used for samples were prepared from surface chemically pure 50 mM sodium phosphate buffered water, pH 7.0.

**Figure 9 nanomaterials-11-01612-f009:**
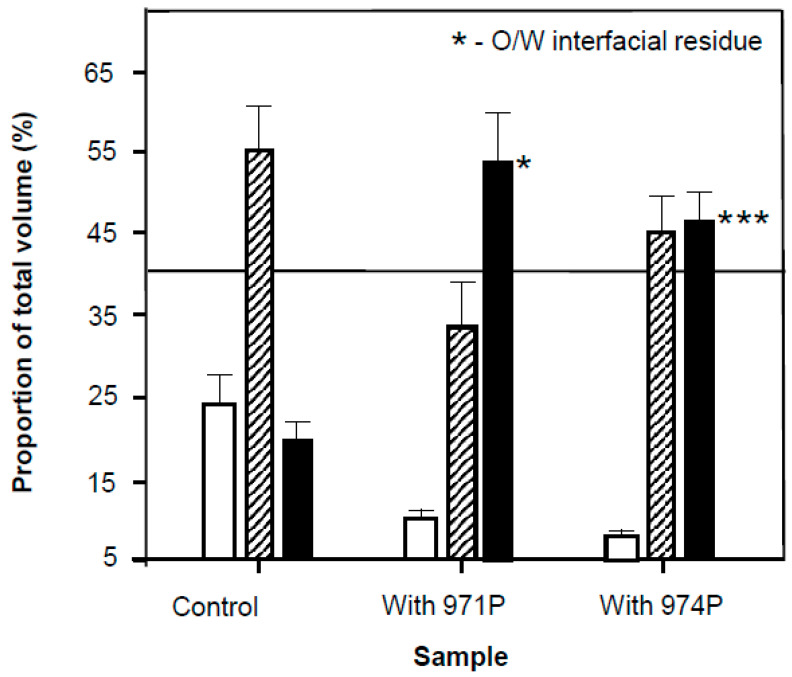
The proportion of various phases of preformed emulsions in the absence and in the presence of carbopol (test system) in the aqueous phase at 20 °C in the presence of 1 × 10^−5^ M sodium azide. The control sample only contained 1 mM Tween and 0.65 mM betamethasone 21-phosphate, whereas the other samples contain 0.5% *w*/*v* carbopol 971P and carbopol 974P. The relative proportion as part of the total sample of oil (unfilled bars), discernible intact emulsion (hatched bars) and aqueous portion with some degree of opacity (solid bars) are presented. The horizontal line is intended as an aid to the eye and indicates the proportion of oil incorporated into the emulsion. Both carbopol containing samples showed evidence of interfacial phase separation but notably the carbopol 974P sample showed evidence on inspection more than 10 min after preparation of a clearly visible 1–2 mm interface ‘sludge’. The sludge is presumed to consist of droplets and polymer chains of sufficient colloidal dimensions to remain suspended. All solutions used for samples were prepared from surface chemically pure 50 mM sodium phosphate buffered water, pH 7.0. Asterisks indicated in the plot point to the amount of interfacial residue (*-slight residue; ***-triple the volume of residue seen in the former).

**Figure 10 nanomaterials-11-01612-f010:**
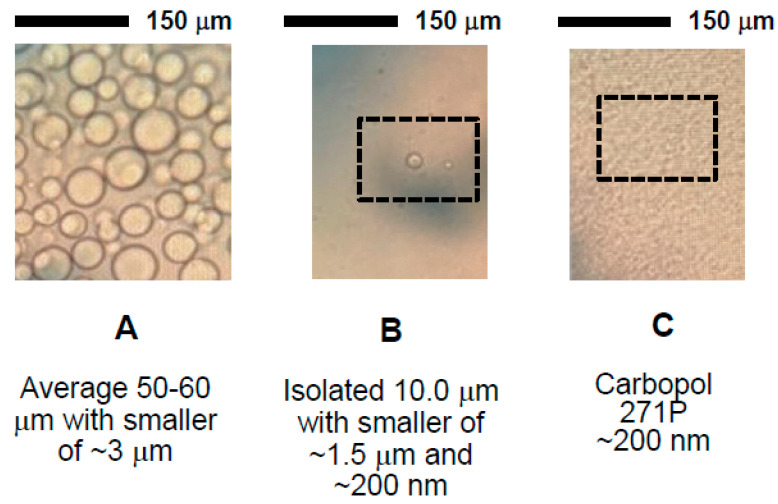
Light microscope images of emulsions formed from 1 mM Tween 80, 0.65 mM betamethasone 21-phosphate and 0.5% *w*/*v* carbopol 971P, immediately after formation with large droplets averaging approximately 60 μm (**A**), a portion of the subnatant after 10 min where with oil droplets ranging from 10 to 1.5 μm and also some particles estimated to be approximately 200 nm (**B**). The suspended material from a dispersion of 0.5% *w*/*v* carbopol 971P was also sampled after 10 min and is indicated the presence of nanoparticles (**C**). All solutions used for samples were prepared from surface chemically pure 50 mM sodium phosphate buffered water, pH 7.0.

**Figure 11 nanomaterials-11-01612-f011:**
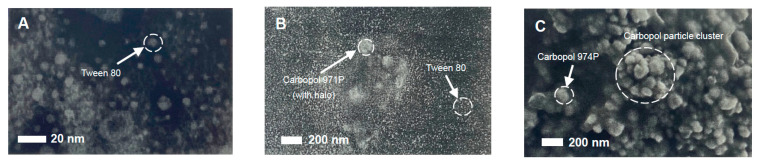
Scanning electron microscopy (SEM) images at an operating acceleration voltage of 2–5 keV. SEM images of pure 1 mM Tween 80 samples (**A**) and also 1 mM Tween 80 in the presence of a low loading of 0.1% *w*/*v* dispersions of carbopol 971P (**B**) and a higher loading of 2% *w*/*v* dispersions of carbopol 974P (**C**). Samples that included carbopols showed the presence of large 100–200 nm particles (images B and C), which associated into even larger aggregates (>500 nm), as shown in the SEM image at the higher concentration for carbopol 974P. All solutions used for samples were prepared from surface chemically pure 50 mM sodium phosphate buffered water, pH 7.0. Images are not all on the same scale.

**Figure 12 nanomaterials-11-01612-f012:**
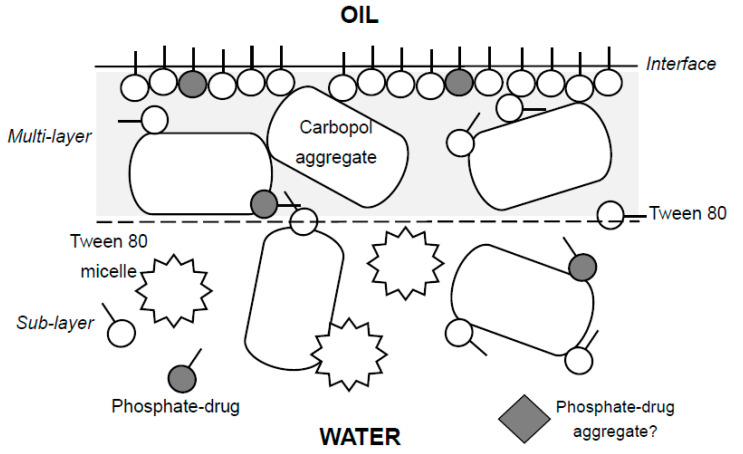
Phenomenological cartoon showing the representation of the various ‘elements’ making up the interfacial adsorbed layer associated with Tween 80, betamethasone 21-phosphate (BMP) and carbopol based oil-in-water emulsions. Key elements are the hydrogen bonded associate carbopol-Tween interfacial adsorbate, free carbopol and free Tween 80 along with micelles of Tween 80. The BMP drug may also associate into aggregates but this is surmised from the surface tension plot (Figure 5). Other associations may include apolar interaction between the alkyl tails of the primary surfactant, with each other and also with the BMP.

**Table 1 nanomaterials-11-01612-t001:** The average size of droplets in O/W batches of emulsions (*n* = 3) formed from 1 mM Tween 80 solutions in the absence and presence of betamethasone phosphate (BMP) and fabricated at 25 °C in buffered media at pH 7.0.

Sample Emulsion	Incorporated BMP (mM)	Diameter, *D*_4,3_ (μm)
Tween alone	-	55.1 ± 0.96
Tween-BMP1	0.10	55.3 ± 0.83
Tween-BMP2	0.25	54.7 ± 0.75
Tween-BMP3	0.45	55.5 ± 1.01
Tween-BMP4	0.65	55.2 ± 0.77
Tween-BMP5	0.80	55.8 ± 1.54
Tween-BMP6	1.00	57.0 ± 1.92

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
