# Peer review of "Interfacial Effects and the Nano-Scale Disruption in Adsorbed-Layer of Acrylate Polymer-Tween 80 Fabricated Steroid-Bearing Emulsions: A Rheological Study of Supramolecular Materials"

_nanomaterials, 2021, doi:10.3390/nano11061612_

Round 1

Reviewer 1 Report

The paper is interesting, and in my opinion suitable for publication, after some changes have been done.

Viscoelasticity was measured by running amplitude sweep and frequency sweep test, but no data are reported a part for an apparent viscosity diagram in Figure 6.
I would encourage including amplitude sweep data, i.e. G’, G’’ as function of stress, at given frequency (if not relevant at least in the supplementary), and most of all frequency sweep analysis with G’, G’’, and eta*, at constant stress as function of frequency. I would also report a flow curve with shear viscosity as function of shear rate (if available).

Please clarify how was determined the error bar in drop size measurement. Is this the error provided by the instrument, or the standard deviation of independent measurements? If the last, by independent measurement is intended independent formulations, or reproduced measure of the same sample?

Detailed analysis of possible source of error in the characterization of droplet size distribution, with specific focus on coalescence phenomena, was previously widely analyzed in the literature. See for example Caserta et al. Rheologica Acta 2004 and 2006.

Author Response

Details to address the queries are provided in the attached *.doxc document

Reviewer 2 Report

The submitted manuscript is interesting, well prepared and and the reported results are well presented and discussed. Some recommendations are below listed:

  • The state of the art reported in the introduction needs to be updated by adding more references related to the use of nano&microcarriers in the targeted field. Fessi's et all have described various aspects in the field of drug delivery of all carriers.
  • What is the exact raison of using Carbopol in the formulation of emulsion.
  • If the size of droplets is measured after dilution, then the dilution may affect the size. Normally, the size should be performed by keeping the same continuous phase (supernatant) composition of mother emulsion.
  • The word nanoparticles should be removed from the manuscript since all sizes are above micrometer. 
  • Any possible interactions between Tween and carbopols? This wasn't discussed in the manuscript. 
  • The same question between Tween and betamethasone 21-phosphate ?

Author Response

Responses to the the raised points are provided in the *.docx document 

Round 2

Reviewer 2 Report

Dear Editor,

The submitted revised version is good and all comments have been considered. Then I have no more suggestions.

Best regards